# V-Cornea: A computational model of corneal epithelium homeostasis, injury, and recovery

**Joel Vanin**[1], **Michael Getz**[1], **Catherine Mahony**[2], **Thomas B. Knudsen**[1], **James A. Glazier**[1]*

**1** Department of Intelligent Systems Engineering and Biocomplexity Institute, Indiana University, Bloomington, Indiana, United States of America, **2** Procter & Gamble Technical Centre, Reading, United Kingdom

\* jaglazier@gmail.com

## Abstract

### Purpose

To develop a computational model that addresses limitations in current ocular irritation assessment methods, particularly regarding long-term effects, and recovery patterns following chemical exposure or trauma to the cornea.

### Methods

V-Cornea is an agent-based computer simulation implemented in CompuCell3D that models corneal epithelial homeostasis and injury response. The model incorporates biologically-inspired rules governing cell behaviors (proliferation, differentiation, death) and key signaling pathways including Epidermal Growth Factor (EGF), translating cell-level behaviors to tissue-level outcomes (*in vitro* to *in vivo* extrapolation, IVIVE).

### Results

V-Cornea successfully reproduces corneal epithelial architecture and maintains tissue homeostasis over extended simulated periods. Following simulated trauma or toxicant exposure, the model accurately predicts healing timeframes of 3–5 days for slight and mild injuries. For moderate injuries with basement membrane disruption, the model demonstrates longer recovery times and emergent dynamic structural disorganization that mimics recurrent corneal erosions, providing mechanistic insights into these pathological conditions.

### Conclusions

V-Cornea's modular CompuCell3D implementation is easily extensible to incorporate additional recovery and injury mechanisms. Future versions will include more realistic basement membrane dynamics and explicit representation of stromal tissue and

**Data availability statement:** All model code, simulation files, scripts for plots, and data used to generate the results presented in this study are openly available in a public repository. The V-Cornea model, implemented in CompuCell3D, along with the scripts for the plots and the data files, can be accessed on GitHub at https://github.com/VaninJoel/vCornea. Additionally, a permanent, versioned archive of the code is also available on Zenodo at https://doi.org/10.5281/zenodo.16764318. All other data supporting the findings of this study are available within the article and its Supporting information files.

**Funding:** This work was supported by Procter & Gamble through a research agreement with Indiana University (salary support to JV, TBK, JAG, and MG). CM is an employee of Procter & Gamble. JAG and MG received additional support from the National Science Foundation (NSF 2303695 to JAG; NSF 2120200 to JAG and MG) and from the National Institutes of Health (NIH U24 EB028887 to JAG). The funders had no role in study design, data collection and analysis, decision to publish, or preparation of the manuscript. Procter & Gamble reviewed the final manuscript for compliance with internal policies, but did not influence the interpretation of the results or the authors' decision to submit the work for publication, nor the preparation and contents of the manuscript.

**Competing interests:** I have read the journal's policy and the authors of this manuscript have the following competing interests: CM is an employee of Procter & Gamble, which funded this research. This work was supported by Procter & Gamble through a research agreement with Indiana University. The other authors declare that they have no competing interests. This does not alter our adherence to the journal's policies on sharing data and materials.

immune response, to improve predictivity for moderate injuries. This virtual-tissue approach shows potential not only for toxicological assessments but also for drug discovery and therapy optimization by providing a platform to test interventions and predict outcomes across various injury scenarios.

## Author summary

Injuries to the cornea are a common health concern, typically resulting from chemical exposure, physical trauma, infection, or environmental damage. Current methods for assessing eye irritation often rely on animal testing or simplified laboratory assays that provide only static snapshots of tissue damage. Clinical time series data on recovery are limited, making predicting long-term healing outcomes and distinguishing between superficial injuries (completely healed) versus deeper injuries (with persistent complications) difficult. To address these limitations in understanding and predicting corneal recovery, we developed V-Cornea, a computer simulation of the human corneal epithelium. By programming individual virtual cells with biologically-motivated rules for growth, movement, differentiation, and death, our model accurately builds and maintains the tissue's layered structure and continuous 7–14 day renewal cycle. Crucially, the simulation mimics clinical healing patterns: superficial injuries repair fully within 3–5 days, while deeper injuries compromising the basement membrane lead to incomplete healing and instability resembling recurrent corneal erosion. This virtual tissue approach demonstrates how microscopic cellular mechanisms drive tissue-level recovery in the cornea. V-Cornea offers a flexible, animal-free platform for investigating chemical toxicity mechanisms and optimizing therapeutic interventions for corneal injuries, providing mechanistic insights into wound healing that are unattainable through traditional experimental assays.

## 1. Introduction

Toxicological assessment, drug discovery, and therapy optimization in the cornea all require the prediction of systemic outcomes from molecular perturbations in the cornea. Accidental exposures to consumer products and industrial chemicals cause eye irritation and corneal damage creating a significant public health concern. These exposures lead to outcomes ranging from temporary discomfort to permanent vision impairment. We assess ocular irritation potential to ensure the safety of consumer products like shampoos, household cleaning solutions, and laundry detergents, as well as professional products [1]. Similarly, developing effective treatments for corneal disorders such as dry eye disease, keratitis, and corneal dystrophies requires understanding how therapeutic compounds interact with corneal tissue at multiple scales. In both toxicology and therapeutics, current approaches often fail to capture

the full complexity of molecular interactions and their translation to tissue-level effects, limiting our ability to predict outcomes accurately without extensive animal testing.

A chemical's inherent properties, exposure conditions, and biological responses influence the severity and persistence of corneal injury following chemical exposure. Assessment of corneal damage found depth of injury to have an important, if not primary, role in eye irritation [2,3]. Depth of injury, rather than the type of chemical or the mode of chemical damage, is particularly significant for recovery time and thus classification of the toxicant. Epithelial injuries typically heal within 3–5 days (mild injury), while injuries affecting the basement membrane and stroma result in prolonged healing times (21 + days) and potential scarring (moderate to severe injury) [4–7].

Toxicologists evaluate corneal damage after toxicant exposure using a variety of experimental methods. Organotypic *ex vivo* assays such as the Bovine Corneal Opacity and Permeability (BCOP) assay and the Isolated Chicken Eye (ICE) test measure opacity, permeability, swelling, and histopathological changes after chemical exposure [8,9]. However, *ex vivo* models have limitations as they lack vascular circulation and can only be maintained for short periods. In addition, inter-species differences reduce their ability to predict human outcomes accurately [10,11]. Similarly, *in vitro* assays such as the Reconstructed 3D Human Cornea-Like Epithelium (RhCE) test and the Short Time Exposure (STE) test quantify cytotoxicity and epithelial barrier function but fail to capture the complexity of the *in vivo* cornea, including inflammation responses and wound healing [12,13].

To support *in vitro* to *in vivo* extrapolation (IVIVE), we introduce V-Cornea, an agent-based computational model implemented in CompuCell3D [14]. V-Cornea's Virtual Tissue (VT) computer simulations aim to bridge the gap between experimental assays and human outcomes by integrating data from *ex vivo*, *in vitro*, and cross-species tests to predict human corneal damage and recovery after chemical injury. V-Cornea replicates critical behaviors and interactions of specific tissue components across multiple biological scales, integrating molecular interactions with cellular and tissue-level processes [15,16]. When we combine these approaches, our VT simulations complement existing non-animal methods by providing mechanistic insights into how cellular responses in simplified test systems translate to tissue-level outcomes, including the temporal dynamics of injury and repair. While our focus here is on damage and recovery for toxicology, the same model structure can be applied to drug discovery, tissue engineering, and therapy optimization [17–22].

## 1.1. Current approaches in corneal modeling

Computer simulations of the cornea typically address distinct biomedical questions following one of three main approaches: (I) finite element analysis (FEA) for understanding mechanical properties, usually representing the cornea as continuous tissue with elastic/viscoelastic properties, focusing on stress-strain relationships without cellular components; (II) mathematical models using partial differential equations (PDEs) to predict wound healing after chemical or abrasive injury, typically representing cell populations as density fields with migration and proliferation rates; and (III) cellular-based models focusing on tissue maintenance.

Researchers using FEA approaches have gained insights into corneal shape changes after refractive surgery, stress distributions in keratoconus, and deformation under varying intraocular pressures [23,24]. Over time, adding more realistic material properties to these methods—such as nonlinear elasticity and anisotropic collagen alignment—improved the fidelity of simulated mechanical responses [25]. However, corneal FEA-based models define tissues through discretized mesh elements with mechanical constitutive relationships, rather than biological processes governing cell turnover and wound repair. Although some advanced FEA models in other tissues, such as cardiac models [26], do include continuum representations of biological processes like cell turnover and tissue remodeling, to our knowledge, existing corneal FEA models have primarily focused on mechanical properties, rather than the biological dynamics of epithelial maintenance and repair that are central to our investigation.

Partial differential equation models address specific aspects of corneal wound healing, such as epithelial cell migration patterns after abrasion and oxygen distribution during hypoxic conditions [27,28]. More recently, mathematical modeling

approaches using ordinary differential equations (ODEs) have been applied to understand corneal epithelium maintenance. Moraki et al. (2018) [29] developed a model based on chemical master equations, but primarily analyzed the derived ODEs to determine constraints on stem cell numbers, division rates, and generation capacity required for maintaining corneal homeostasis. Their model focused on proliferation kinetics and analyzed factors influencing cell populations at steady state, but did not incorporate spatial dynamics or injury response mechanisms. Multicellular approaches have investigated how different parameters (replicative lifespan, spatial correlations, and bias) affect the overall tissue homeostasis and pattern formation in corneal basal layer, without explicitly modeling biochemical signaling pathways [30]. By simulating individual cells, multicellular models reveal how local cell replication and removal drive tissue-level organization.

Researchers rarely integrate mechanical, chemical, and cellular aspects of corneal biology into a single platform that simulates both normal tissue maintenance and injury response. Our Virtual Cornea (V-Cornea) combines mechanistic rules for individual cell behaviors with a simplified representation of chemical signaling and tissue architecture. This design yields a single platform that unifies cellular, mechanical, and biochemical aspects for studying corneal injury and recovery dynamics.

### 1.2. Biological background

#### 1.2.1. Overview of the corneal system.
The cornea forms the transparent anterior surface of the eye, serving as both a protective barrier and the primary refractive element of the visual system (Fig 1). As the outermost tissue, it

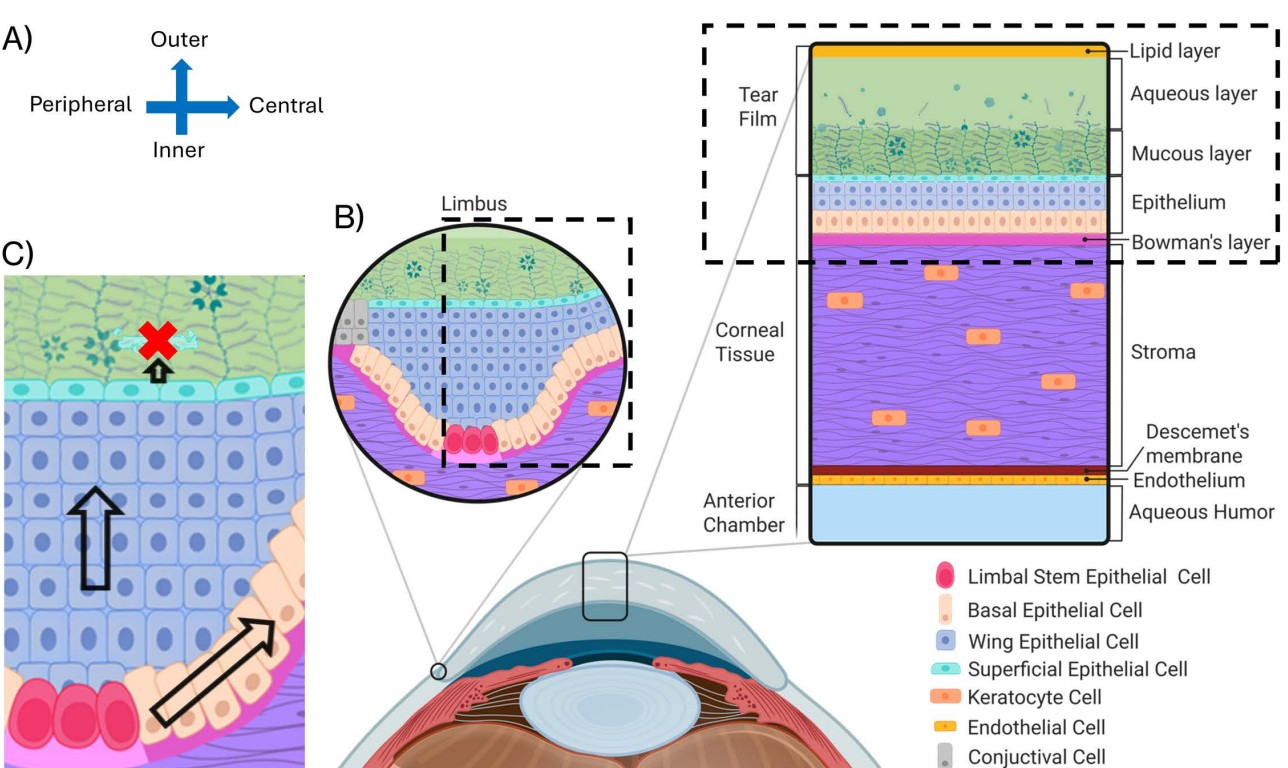

**Fig 1. Corneal epithelium microanatomy and cellular dynamics. (A)** Coordinate system definition: x-axis represents radial position (Peripheral to Central) and y-axis represents depth (Inner to Outer). **(B)** Cross-sectional view of corneal anatomy including the tear film, stroma, and endothelium. The dashed boxes indicate the simulation domain, which focuses on the epithelial and stromal interface and the limbal region of the cornea. **(C)** The Limbal Epithelial Stem Cell (LESC) hypothesis: LESCs (rose-red) in the limbus divide and migrate centripetally (arrows) while differentiating and moving superficially. Created in BioRender. Vanin, J. (2026) https://BioRender.com/lom8bmf.

interfaces directly with the external environment while maintaining optical clarity essential for vision. Structurally, the cornea consists of five distinct layers: the epithelium, Bowman's layer, stroma, Descemet's membrane, and endothelium. Each layer contributes uniquely to corneal function, with the epithelium serving as the primary protective barrier against external threats [31].

The corneal epithelium constitutes approximately 10% of the total corneal thickness (50–52 μm thick) and forms the outermost cellular layer, situated beneath the tear film. This stratified epithelial tissue comprises 4–8 cell layers that form a tightly regulated barrier [31]. The epithelium interfaces with the tear film at its apical surface and with Bowman's layer at its basal surface. The tear film, with its lipid, aqueous, and mucin layers, both protects the cornea from external insults and delivers growth factors such as EGF that regulate epithelial proliferation and differentiation [32–35].

Between the epithelium and Bowman's layer lies the epithelial basement membrane (EpBM) (0.05-0.1 μm thick), a specialized extracellular matrix, which serves as both an anchoring structure for epithelial cells and a selective barrier that regulates molecular exchange [36,37]. The EpBM plays a crucial role in regulating epithelial-stromal interactions during wound healing by controlling cytokine exchange. Wound healing outcomes are significantly affected when damage disrupts the EpBM, because the normal differentiation and adhesion processes are compromised [38–44].

**1.2.2. Epithelial cell types and organization.** The corneal epithelium is organized into a spatially and functionally distinct hierarchy of cell types, each with specific morphological characteristics and functional roles. These cell populations are maintained through a continuous process of renewal originating from stem cells at the corneal periphery. The epithelium comprises four primary cell types arranged in distinct spatial layers:

*Limbal Epithelial Stem Cells* (LESCs) [Fig 1 - rose-red cells] reside at the limbus—the border between cornea and sclera—and provide the cellular source for epithelial renewal. These stem cells are characterized by their high nuclear-to-cytoplasmic ratio and slow-cycling. LESCs require contact with the limbal basement membrane to maintain their stem cell characteristics [45]. They reside within specialized niches in the limbal epithelial crypts, where they receive protection and regulatory signals. These cells retain their proliferative capacity throughout life and serve as the ultimate progenitors for all corneal epithelial cells [46,47].

*Basal Cells* [Fig 1 - peach-orange cells] form a single layer of columnar cells that anchor to the basement membrane via hemidesmosomes and retain proliferative capacity. These cells arise from the differentiation of LESCs as they migrate centripetally away from the limbus. Basal cells maintain regular contact with the basement membrane, which is essential for preserving their proliferative potential [48]. They typically exhibit a polygonal morphology when viewed from the apical surface and feature prominent cell-cell junctions. Basal cells actively participate in both normal epithelial renewal and wound healing responses, with their adhesion to the basement membrane playing a critical role in epithelial integrity.

*Wing Cells* [Fig 1 - blue cells] occupy the intermediate layers (2–3 layers) of the epithelium and derive their name from their characteristic wing-like appearance in cross-section. They arise from the vertical migration and differentiation of basal cells. Wing cells feature a more flattened morphology than basal cells, but are not yet fully differentiated. These cells serve as transitional elements that facilitate basal cell movement toward the corneal surface while maintaining epithelial structural integrity [49]. Though not proliferative under normal conditions, wing cells retain some capacity to respond to injury signals.

*Superficial Cells* [Fig 1 - cyan cells] form the outermost 1–2 layers of the epithelium and represent the terminal stage of epithelial differentiation. These squamous cells are characterized by their remarkably flattened morphology and specialized apical microvilli, which help stabilize the tear film. The most distinctive feature of superficial cells is their formation of tight junctions with a restrictive pore radius of approximately 4 Å [50]. These junctions create a highly selective barrier that regulates fluid and molecular exchange between the tear film and underlying corneal tissue. Superficial cells exhibit a finite lifespan of approximately 7–10 days before undergoing a specialized form of non-classic apoptosis and desquamation from the corneal surface [41,51,52].

### 1.2.3. Epithelial cell dynamics and behaviors.

The corneal epithelium maintains its structural integrity and functional capabilities through tightly coordinated cellular behaviors including proliferation, differentiation, migration, and regulated cell death. These dynamic processes work in concert to ensure continuous renewal while preserving the epithelium's barrier function.

**Proliferation and Growth Regulation**

Epithelial renewal begins with the proliferation of limbal epithelial stem cells, which exhibit a specialized centripetal division pattern [53]. This directional division produces progenitor cells that migrate toward the central cornea, a process crucial for replenishing the central epithelium and preserving corneal transparency [53]. LESC proliferation is precisely regulated by intrinsic genetic programs and extrinsic signaling from the surrounding niche environment [47].

Basal cells also contribute significantly to epithelial renewal through their proliferative capacity. Unlike the directed division pattern of LESCs, basal cells follow a random cleavage plane orientation during division. This randomized pattern serves a crucial role in maintaining tissue homeostasis by promoting even cell distribution [48].

Growth factor signaling plays a critical role in regulating epithelial proliferation, with Epidermal Growth Factor (EGF) serving as a primary mitogenic signal [54]. EGF stimulates corneal epithelial proliferation particularly during wound healing, while its effect on intact epithelium remains minimal. Interestingly, even with continued EGF treatment, any hyperplastic response eventually returns to normal epithelial thickness, demonstrating the presence of intrinsic regulatory mechanisms [55]. The tight junctions formed by superficial cells help regulate EGF penetration into the epithelium with their restrictive pore radius (approximately 4 Å) [50] limiting the diffusion of larger EGF molecules (~6 kDa) [56] under normal conditions.

**Differentiation**

Epithelial differentiation follows a precisely orchestrated sequence that begins with LESCs and culminates in terminally differentiated superficial cells. This process is primarily regulated by spatial cues and cell-matrix interactions. LESC differentiation into basal cells is triggered when stem cells lose contact with the specialized limbal basement membrane [45,57]. This spatial signal initiates the first step in the differentiation cascade as cells migrate centripetally.

Basal cells differentiate into wing cells when they detach from the basement membrane and move vertically within the epithelium. This transition involves significant cytoskeletal reorganization and changes in cell-cell adhesion properties [48].

Wing cells undergo terminal differentiation into superficial cells as they approach the epithelial surface. This final differentiation step involves dramatic morphological changes, including flattening of the cell shape, formation of specialized apical structures, and assembly of tight junctional complexes [49,58].

Throughout this differentiation sequence, cells undergo progressive gene expression changes that reflect their specialized functions at each stage. The entire renewal process, from stem cell division to superficial cell desquamation, takes approximately 7–14 days [59,60], establishing a continuous cycle that maintains epithelial homeostasis.

**Migration**

Cell migration in the corneal epithelium follows distinct patterns that contribute to tissue maintenance and repair.

*Centripetal migration* – following division, LESCs` LESCs and their progeny migrate centripetally from the limbus toward the central cornea. This directional movement establishes a continuous flow of new cells to replace those shed from the central epithelium. This migration is regulated by a complex network of adhesion molecules and extracellular matrix interactions. Specifically, proteins such as laminin and fibronectin provide guidance cues and adhesion substrates that facilitate this directed cell movement [61,62].

*Vertical migration* – as cells differentiate, they simultaneously migrate vertically from the basal layer toward the corneal surface. This upward movement is facilitated by a combination of cell proliferation in the basal layer, which provides physical impetus, and changes in cellular adhesion properties during differentiation [58].

*Wound healing migration* – following epithelial injury, basal cells serve as the primary cells that migrate into the wound area [63]. This directed movement originates from increased cell proliferation in the limbal region and altered adhesion dynamics, with mechanical constraints from neighboring cells helping create movement toward the wound site [64,65].

**Cell Death and Shedding**

Superficial cells undergo a specialized form of programmed cell death before being shed from the corneal surface. This process differs from classical apoptosis and is designed to maintain barrier integrity even as individual cells are removed from the epithelium [52]. The controlled desquamation of superficial cells completes the epithelial renewal cycle and is finely balanced with the production of new cells from the basal layers. This balance ensures that the epithelium maintains consistent thickness and structural organization while continuously renewing its cellular components.

**1.2.4. Epithelial response to injury.** Following injury, tear production increases as part of the physiological response [66]. Damage to the superficial cell layer allows increased influx of growth factors from the tear film, triggering repair responses [5,66,67]. Injury-induced breaches in the epithelial barrier allow increased EGF diffusion into the tissue triggering basal cells to migrate toward the wound site. The healing process progresses through a sequence of basal cell coverage over the wound bed, followed by cell differentiation to restore proper epithelial stratification [63]. The severity and nature of the injury, particularly whether it affects the basement membrane, significantly influence the healing outcome [43,68–70]. When the basement membrane remains intact, epithelial healing typically proceeds rapidly (3–5 days) and without complications [7,71]. However, when the basement membrane is damaged, healing may be delayed, and the risk of recurrent erosions increases due to impaired epithelial adhesion [72,73].

## 1.3. Scope of the V-cornea model

Our current implementation of V-Cornea focuses specifically on the epithelium, tear film, and a simplified representation of the basement membrane, as these structures are directly involved in the initial response to chemical exposure as well as mild to moderate injuries [2,3,6]. Since the stroma is minimally represented in our model as a spatial constraint, we have not implemented its complex cellular dynamics or collagen organization in this version. [74–76]

We have intentionally excluded detailed representations of the deeper corneal structures (Descemet's membrane and endothelium [77,78]) in this initial model, as they typically become involved only in severe injury scenarios. Additionally, we have simplified the immune response aspects of corneal wound healing to focus first on the fundamental epithelial regeneration processes [79,80]. These simplifications allow us to create a computationally tractable model that accurately captures the essential dynamics of epithelial homeostasis, as well as both successful recovery after slight to mild injuries and failed recovery after moderate injuries, which are the most common outcomes in accidental chemical exposures. Future iterations of V-Cornea will incorporate these additional structures and processes as we extend the model to simulate more severe injury scenarios.

In this work, we introduce the Virtual Cornea (V-Cornea), a two-dimensional agent-based model (ABM) to elucidate key mechanisms underlying corneal epithelial homeostasis and injury response. While organotypic assays and RhCE tests measure cell death and barrier disruption as endpoints, they fail to capture longitudinal dynamic interactions between cell behaviors and tissue-level organization during injury and recovery. We begin to address this gap by implementing biologically-inspired rules for cell proliferation, differentiation, death, and migration to predict how cell-level responses create tissue-level outcomes. Our mechanistic approach reveals how initial cell damage patterns lead to different times to recovery and different spatial patterns during recovery. For moderate injuries we observe behavior mimicking recurrent corneal erosions following basement-membrane disruption. We can simulate extended time periods (up to 6 months) to investigate long-term tissue responses that researchers cannot practically study *ex vivo*, making our model, in the long term, particularly valuable for understanding how different initial cell injury patterns lead to distinct adverse outcomes.

## 2. Methods and implementation

### 2.1. Modeling approach

We implemented V-Cornea using agent-based modeling (ABM) to represent the spatiotemporal dynamics of corneal epithelial cells. This approach allows individual cells to act as autonomous agents governed by programmed rules [Fig 2],

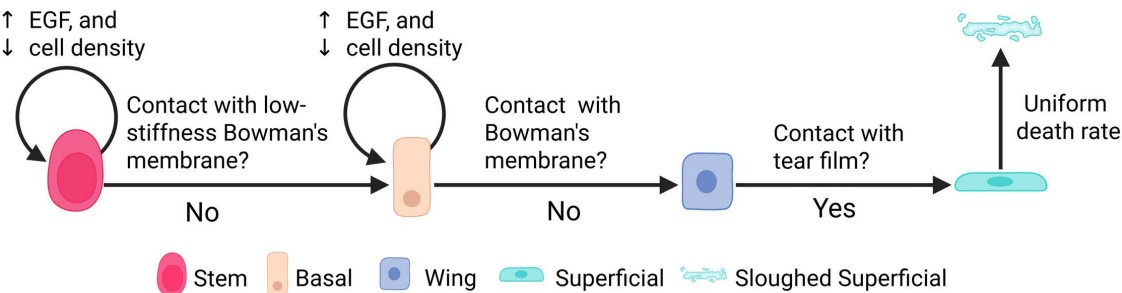

**Fig 2. Corneal epithelial cell state transitions and regulatory mechanisms.** Modeled life-cycle states include Stem (rose-red), Basal (peach-orange), Wing (blue), and Superficial (cyan). Proliferation of Stem and Basal cells is upregulated by EGF and downregulated by cell density (curved arrows). Transitions occur upon loss of contact with specific regions: Stem to Basal (Limbal EpBM/Bowman's membrane complex), Basal to Wing (EpBM/Bowman's membrane complex), and Wing to Superficial (tear film contact). Superficial cells undergo probabilistic sloughing to maintain physiological turnover. Created in BioRender. Vanin, J. (2026) https://BioRender.com/t4tk7sd.

interacting with neighboring cells and their microenvironment. We selected ABM because it effectively captures the coordinated cell behaviors essential to corneal epithelial homeostasis and wound healing, which are difficult to model using continuum methods.

Our implementation uses CompuCell3D (version 4.6.0), an open-source, cross-platform environment designed for simulating multi-cellular dynamics (https://compucell3d.org) [14]. CompuCell3D employs the Glazier-Graner-Hogeweg (GGH) formalism [81], which represents cells as collections of lattice sites and evolves them according to effective-energy minimization principles. The ABM approach has demonstrated success in modeling diverse multicellular processes [82] and enables us to represent key cellular mechanisms and the spatially-resolved chemical environment critical for our simulation.

## 2.2. Simulation domain and spatial framework

Our model represents a two-dimensional (2D) radial sagittal section of the corneal limbus and peripheral cornea. The simulation domain is implemented as a discrete square lattice of 200×90 voxels [Fig 3], where each voxel corresponds to a 2×2 µm area (4 µm²), resulting in a total simulated region of 400×180 µm. Within this domain, the limbal region spans approximately 80 µm in width, while the peripheral corneal region extends a further ~320 µm toward the central cornea. We restrict the domain to the limbus and adjacent peripheral cornea rather than the entire cornea for computational efficiency. Additionally, this region contains the limbal stem cell niche and captures the key epithelial dynamics of interest. The modeled epithelial thickness is ~50–52 µm, corresponding to ~4–8 cell layers, which falls within the range of central epithelial thickness reported for healthy human corneas by SD-OCT and AS-OCT imaging (≈48–55 µm) [83]. Limbal epithelial thickness measurements in normal eyes are typically around 80–85 µm [84], substantially thicker than the central epithelium. Our simulated limbal region is correspondingly thicker than the central epithelium, consistent with the limbal–central gradient observed *in vivo*. We modeled epithelial cell diameters ranging from 10–30 µm [85] and implemented a simulated tear film layer of 5–10 µm to reflect physiological conditions.

Geometrically, the *in vivo* cornea is part of a curved 3D shell with an average anterior radius of curvature of approximately 7.5–8.0 mm [86]. Our 2D model approximates this shell by a flat rectangular domain along one meridian. Because the simulated radial extent (400 µm) is small relative to the corneal radius (~8 mm), the geometric distortion caused by representing the curved surface as a flat 2D plane is negligible. Thus, over the spatial scales we consider, flattening the curved shell introduces only small errors in path lengths and local interface geometry for migrating cells, relative to biological variability in epithelial thickness, cell size, and motility.

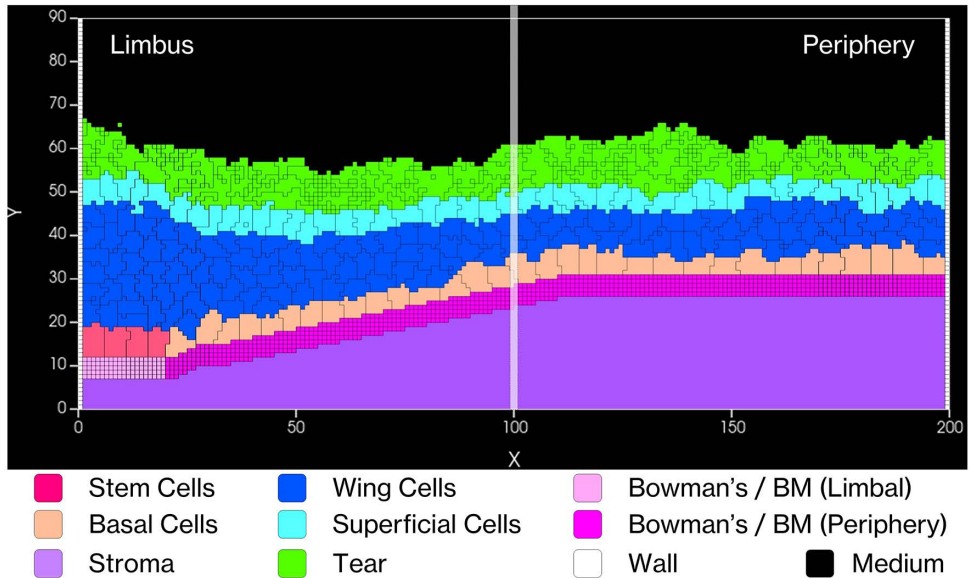

**Fig 3. Simulated corneal epithelium structure.** V-Cornea CompuCell3D rendering (**200** × **90** voxels). The model represents the Stroma (lilac-purple), Bowman's layer/EpBM (pink/magenta), Limbal Stem Cells (rose-red), Basal (peach-orange), Wing (blue), and Superficial cells (cyan) topped by Tear film (green). The medium (black) represents air.

A more important consequence of the 2D representation is how it relates to the full 3D tissue. Each point in our 2D radial slice corresponds, *in vivo*, to a ring of tissue that extends around the corneal circumference at that radius. Quantities such as cell number, wound area, and chemical load in the model should therefore be interpreted as *densities per unit circumferential length* (analogous to "cells per millimeter of corneal circumference") rather than absolute totals for the entire cornea. In a 3D eye with a roughly circular epithelial defect, the true wound area at radius $r$ scales with the circumferential length $\sim 2\pi r$, while our 2D slice describes how far the epithelial front has advanced along one radial line. Provided limbal function is approximately uniform around the circumference, the radial healing kinetics in our slice (front position vs. time and depth) are expected to match the local contribution to global wound closure, even though the model does not explicitly track the full plan-view wound shape.

This interpretation is consistent with clinical imaging of large epithelial abrasions. Dua and Forrester [87] showed that re-epithelialization in patients follows a reproducible pattern: three to six convex epithelial fronts develop along the wound circumference and progress toward the center, neighboring fronts meet to form polygonal contact geometries, and the final closure often exhibits a characteristic Y-shaped contact line. The area of the epithelial defect decreases approximately exponentially with time, indicating an almost constant rate of epithelial migration along the wound edge. In this regime, each radial meridian across the defect effectively experiences a locally advancing front with similar speed. Our 2D radial model is designed to capture exactly this radial component of healing—how an epithelial front advances with depth and radial position—rather than the global plan-view geometry of multiple fronts and their circumferential interactions.

The dimensional reduction from 3D to 2D also affects mechanics and cell rearrangements. In the real cornea, epithelial cells and the underlying extracellular matrix can deform and rearrange in three dimensions, including circumferentially and out of the sagittal plane. In contrast, cells in our 2D cross-section are confined to a plane: contact interfaces are represented as line segments rather than surface areas, forces are effectively projected into that plane, and cells cannot "move around" each other by stepping into the third dimension. This constraint tends to increase local crowding and jamming at a given cell density and alters how mechanical stresses percolate through the tissue compared to a 3D model. As a result,

we do not interpret parameters such as contact energies or motility coefficients as direct estimates of 3D mechanical quantities; instead, we treat them phenomenologically and calibrate them so that the model reproduces population-level observables such as epithelial thickness profiles, turnover times, clonal patterns, and wound closure times.

These scaling and mechanical considerations also define which questions the current 2D framework can, and cannot, address. V-Cornea is well suited to studying local depth- and radius-dependent behaviors: maintenance of homeostasis, differences between slight (superficial) and mild (full-epithelium) injuries, the consequences of basement membrane damage, and the associated healing time courses when limbal stem cell supply is intact and approximately symmetric. In contrast, phenomena that are fundamentally circumferential or sectoral—such as the emergence of multiple discrete fronts and Y-shaped contact lines in plain view, asymmetric healing when only part of the limbus is functional, or progressive conjunctivalisation creeping across the cornea in limbal stem cell deficiency—lie outside the scope of a single 2D radial slice. Addressing those questions would require at least a surface model defined over radius and angle (r,θ), or a full 3D implementation of the corneal epithelium and conjunctiva.

Within this interpretation, the modest geometric distortions introduced by flattening the cornea and the enhanced crowding inherent to 2D are absorbed into the calibrated model parameters. Consequently, even though individual cell–cell and cell–matrix interactions in the simulation differ quantitatively from those in a full 3D tissue, the emergent tissue-level behaviors we focus on—homeostatic epithelial turnover, classification of injuries by depth, and the contrasting recovery dynamics of epithelial-only versus basement-membrane–disrupting injuries—remain representative of biological observations. The spatial mapping (2 μm per lattice site) and temporal mapping (1 Monte Carlo step ≈ 6 minutes of biological time) provide a direct correspondence between simulation outputs and experimental or clinical measurements of epithelial thickness and wound closure dynamics.

### 2.3. Cellular components and agent representation

**2.3.1. Cell types.** V-Cornea implements four distinct cellular agents representing the corneal epithelial cell types, each with specific properties and behaviors. *Limbal Epithelial Stem Cells* (LESCs) [Fig 4 - rose-red agents] are implemented as proliferative agents that maintain contact with the limbal basement membrane and divide to produce progenitor cells that migrate centripetally [45–47,57]. *Basal Cells* [Fig 4 - peach-orange] are modeled as proliferative cells that maintain contact with the basement membrane, with division following a random cleavage plane orientation to ensure balanced tissue distribution [48]. *Wing Cells* [Fig 4 - blue] function as non-proliferative transitional components in the epithelium [49].

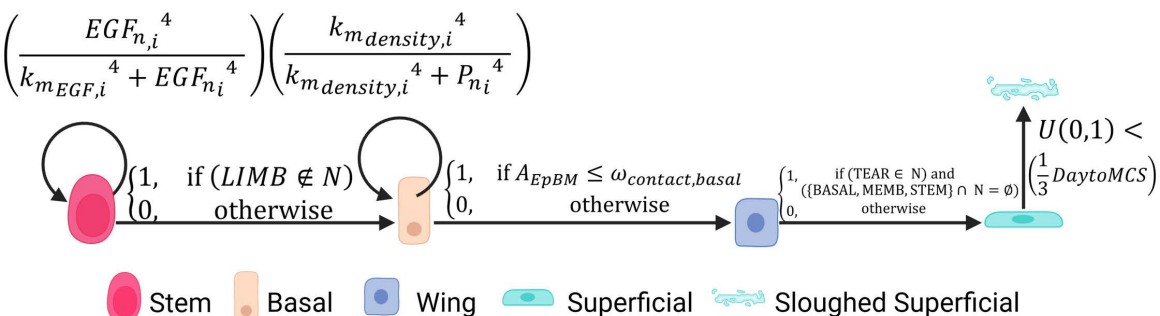

**Fig 4. Mathematical implementation of differentiation and apoptosis rules.** State transitions follow logical conditions based on neighbor interactions (**N**). Stem cells differentiate upon losing Limbal contact ([Eq 4]). Basal cells differentiate when EpBM contact area (**A**$_{EpBM}$) drops below $\omega_{contact,basal}$ ([Eq 5]). Wing cells differentiate upon reaching the Tear film ([Eq 6]). Sloughing follows a uniform probability distribution calibrated to **DaytoMCS** = 240. Created in BioRender. Vanin, J. (2026) https://BioRender.com/pyedehe.

*Superficial Cells* [Fig 4 - cyan] are programmed to form a protective barrier at the apical surface with a finite lifespan, after which they undergo a probabilistic sloughing mechanism [41,51,52].

**2.3.2. Tear and epithelial basement membrane as agents.** Our model incorporates several environmental components critical for epithelial function. The *Tear Film* [Fig 3 - bright green] is implemented with a fixed level of Epidermal Growth Factor (EGF), meaning that EGF is available at all positions along the corneal surface [34,35,54,66]. The *Epithelial Basement Membrane* (EpBM) [Fig 3 - pink/magenta horizontal structure] is modeled as a unified computational entity that combines the biological epithelial basement membrane and Bowman's layer, represented with two distinct regions: low-stiffness limbal regions (pink) and high-stiffness central regions (magenta) [36,37,39,40,45]. The *Stroma* [Fig 3 - purple] is implemented as a simplified solid material that excludes the entry of other cells following injury.

## 2.4. Cellular behaviors and rules

We implemented cell behaviors through a set of programmed rules that govern cell differentiation, proliferation, movement, and death (S1-S5 Tables).

**2.4.1. Growth dynamics.** Cell growth is calculated based on two primary factors: local EGF concentration and mechanical constraints from neighboring cells. Each cell's growth rate is determined by how its target volume progresses over time. The EGF-dependent growth rate is calculated using a Hill function:

$$G_{nEGF,i} = \left( \frac{EGF_{n,i}^4}{k_{mEGF,i}^4 + EGF_{ni}^4} \right),$$

(1)

where $EGF_{ni}$ is the average EGF concentration in cell $i$ at time $n$.

Density-dependent growth inhibition is represented by another Hill function:

$$G_{ndensity,i} = \left( \frac{k_{mdensity,i}^4}{k_{mdensity,i}^4 + P_{ni}^4} \right),$$

(2)

where $P_{ni}$ is the effective "pressure" inside cell $i$ at time $n$, derived from its volume deviation.

In both Hill functions we use an exponent $n = 4$. This value is not meant to represent the stoichiometry of a single molecular interaction, but rather an *effective* ultrasensitive switch between proliferative and quiescent states. Multi-step kinase cascades such as MAPK/ERK, which mediate growth-factor responses, produce stimulus–response curves with apparent Hill coefficients on the order of 4–5 [88], reflecting strong ultrasensitivity and noise filtering in the proliferation decision. Likewise, quantitative measurements of contact inhibition in epithelial monolayers show that the decline of mitotic rate with increasing local density is well-described by a steep Hill function, with exponents near 4 [89]. In preliminary simulations, lower exponents (e.g., $n = 1$–2) produced overly graded, "leaky" proliferation and failed to maintain realistic epithelial thickness, turnover times, and injury-healing dynamics, whereas $n = 4$ allowed us to satisfy these constraints simultaneously.

The total growth rate combines these factors as

$$G_{ntotal,i} = \delta_i \times G_{ndensity,i} \times G_{nEGF,i},$$

(3)

where $\delta_i$ is the cell's intrinsic maximal doubling rate [29]. The parameter values are calibrated to ensure physiologically relevant growth patterns, with detailed formulations provided in S1 Text.

**2.4.2. Mitosis and cell division.** Cell division is implemented using a volume-based regulation system. We initialize both LESCs and basal cells with a volume of 25 pixels (100 μm²) and trigger mitosis when cells reach double their initial volume—50 pixels (200 μm²). During division, the parent cell's volume is equally distributed between the two resulting

cells. We implemented different division patterns for different cell types. For LESCs, we oriented the division plane to direct daughter cells centripetally, following a pattern that maintains stem cell populations while producing committed progenitors [61,62,64]. For basal cells, we implemented a random cleavage plane orientation to promote even tissue distribution throughout the epithelium [48].

**2.4.3. Differentiation.** Cell differentiation is implemented as a rule-based process triggered by spatial relationships. **LESC to Basal** transition occurs when an LESC loses contact with the limbal EpBM [Fig 4 - first rule] [45,57], represented as

$$P_{(STEM \rightarrow BASAL)} = \begin{cases} 1, & \text{if } (LIMB \notin N) \\ 0, & \text{otherwise} \end{cases},$$

(4)

where $N$ is the set of cell types neighboring the cell being evaluated, and the symbol $\notin$ indicates that the **LIMB** cell type is *not* present in that neighborhood.

**Basal to Wing** transition is triggered when contact with the basement membrane diminishes [48] below a defined threshold:

$$P_{(BASAL \rightarrow WING)} = \begin{cases} 1, & \text{if } A_{EpBM} \leq \omega_{contact,basal} \\ 0, & \text{otherwise} \end{cases},$$

(5)

where $A_{EpBM}$ is the number of pixels in contact with the basement membrane, and $\omega_{contact,basal} = 5$ pixels is the minimal contact area required to maintain basal phenotype.

**Wing to Superficial** transition occurs when wing cells reach the uppermost region [49,58] To differentiate, the cell must contact the Tear film **and** must have completely lost contact with the lower layers (Basal, Stem, or Basement Membrane):

$$P_{(WING \rightarrow SUPER)} = \begin{cases} 1, & \text{if } (\{TEAR, \ WING\} \in \ N) \text{ and } (\{BASAL, \ MEMB, STEM\} \cap N = \varnothing) \\ 0, & \text{otherwise} \end{cases}$$

(6)

In this equation, TEAR $\in$ N requires contact with the tear film, while the second condition specifies that the intersection ($\cap$) between the cell's neighbors (N) and the lower-layer cell types must be empty ($\varnothing$). These transitions are deterministic and occur immediately once the conditions are met.

**2.4.4. Movement.** Cell movement is implemented through the Cellular Potts Model framework, where cell positions evolve to minimize the system's Hamiltonian:

$$\Delta H = \Delta H_{contact} + \Delta H_{links} + \Delta H_{volume} + \Delta H_{surface} + \Delta H_{chemotaxis}.$$

(7)

The contact energy term controls adhesion between different cell types, while volume and surface constraint terms maintain cell size and shape. The chemotaxis term enables directed movement in response to chemical gradients. For wound healing scenarios, we implemented directed migration by adjusting chemotactic responses, particularly for basal cells, to move toward the wound site following the basal cell migration hypothesis [63]. Cell positions update through the Monte Carlo algorithm that accepts configuration changes when they decrease the system energy or with a Boltzmann probability when they increase it. (S3 Text)

**2.4.5. Cell death and sloughing.** We implemented two distinct mechanisms for cell removal. Natural sloughing occurs for superficial cells in contact with the tear film through a probabilistic approach:

$$P_{slough} = U(0,1) < \left(\tfrac{1}{3} DaytoMCS\right),$$

(8)

where $U(0,1)$ is a uniform random sample in [0,1], and *DaytoMCS* is the conversion factor between days and Monte Carlo Steps (240). This calibrated probability ensures physiological epithelial turnover rates of 7–14 days [90] with an average

transit time of approximately 1.75 days per cell layer, accommodating the natural variation in epithelial thickness from 4 layers at the periphery to 8 layers in the limbal region.

For injury-induced death, we model tissue injury through both ablation and chemical exposure mechanisms. Ablation simulates physical trauma by removing cells within a defined circular region and replacing them with tear film to reflect the physiological response of increased tear production following injury. Chemical exposure uses a reaction-diffusion model to simulate toxicant spread, implementing both localized exposures through droplet-like gaussian distributions and broader uniform exposure patterns. When cells encounter chemical concentrations above defined thresholds, we trigger programmed cell death through volume reduction.

Both mechanisms generate distinct spatial patterns of injury but differ in implementation—ablation applies instantaneous geometric removal, while chemical injury incorporates spatiotemporal dynamics dependent on diffusion patterns and concentration gradients. We have designed our framework with the flexibility to accommodate more complex chemical and cell behaviors in future iterations. The system can incorporate parameters for chemical reactivity, cellular selectivity, and tear film interactions when experimental calibration data becomes available. This extensibility will enable simulation of diverse chemical exposure scenarios, from caustic agents which cause rapid membrane disruption to surfactants that gradually compromise barrier function in a dose- and duration-dependent manner. Our current implementation serves as a proof-of-concept demonstration of our ability to distinguish between injury classifications based on depth and recovery dynamics, rather than as a direct simulation of specific chemical mechanisms. We provide the technical specifics of this implementation in S4Text.

## 2.5. Tear film and epidermal growth factor (EGF) dynamics

We model EGF as a critical signaling molecule using a reaction-diffusion system with two key parameters: diffusion coefficient and decay rate. While our model explicitly implements EGF, this signaling pathway effectively represents an abstraction of multiple growth factors and signaling mechanisms involved in corneal homeostasis and wound healing, including neural regulation [91], inflammatory mediators [92], and other epithelial growth factors [4]. This simplified representation allows us to capture essential regulatory dynamics without the computational complexity of modeling each signaling pathway individually. The tear film serves as a constant EGF source at the epithelial surface, modeled as a region of fixed concentration to represent continuous lacrimal replenishment. The EGF concentration field $c_{EGF}(x, y, t)$ evolves according to the equation:

$$\frac{\partial c_{EGF}}{\partial t} = D_{EGF}(x, y)\nabla^2 c_{EGF} - k_{d_{EGF}} * c_{EGF} + \Omega_{EGF}(x, y, t), \tag{9}$$

where $D_{EGF}(x, y)$ is the cell-type dependent diffusion coefficient, $k_{d_{EGF}}$ is the global decay rate, and, $\Omega_{EGF}(x, y, t)$ represents the net source/sink terms.

We implement cell-type dependent diffusion rates to reflect the distinct permeability characteristics of each epithelial layer. The diffusion coefficient varies by cell type:

$$D_{EGF}(x, y) = \begin{cases} 20.0, & \text{in superficial cells} \, (D_{EGF,super}) \\ 20.0, & \text{in membrane } EpBM \, (D_{EGF,memb}) \\ 0, & \text{in limbal } EpBM \, (D_{EGF,limb}) \\ D_{global_{EGF}} & \text{otherwise} \end{cases} \tag{10}$$

This variation is biologically significant, as the superficial cell layer creates a highly selective barrier through tight junctions. This barrier property is critical for our injury simulations, as damage to the superficial layer allows increased EGF diffusion to basal and stem cells, triggering proliferation and migration responses essential for wound healing.

 

PLOS Computational Biology

Our model incorporates varying chemical diffusion rates across different tissue compartments, with $D_{global_{EGF}}$ set at 186 voxels²/MCS based on experimental data from GelMA hydrogel experiments [93] adapted to our simulation parameters. The global decay constant for EGF ($k_{d_{EGF}}$) is set to 0.5 per MCS, intentionally higher than the physiological baseline values calculated from EGF's circulating half-life of 42–114 minutes [94]. We selected this elevated decay rate to implicitly incorporate additional processes not explicitly modeled, including cellular uptake, receptor-mediated endocytosis, and sequestration by extracellular matrix components.

We establish boundary conditions to create a physically confined system where EGF cannot escape through lateral boundaries (implemented through wall cells and no-flux conditions), top and bottom boundaries maintain zero concentration, and effective transport is confined to the region between wall cells. This implementation enables us to simulate the dynamic EGF concentration gradients that drive cell behavior during both homeostasis and injury response. As shown in Fig 5, we capture how EGF distribution patterns change dramatically following injury, creating localized regions of elevated concentration at wound sites that trigger cellular responses, and how these patterns normalize as the epithelial barrier is restored during healing (see Table 1 for model summary).

## 3. Results

### 3.1. Emergent behavior and tissue homeostasis

V-Cornea demonstrates emergent behavior as complex corneal epithelial structures develop from basic cellular rules. Fig 6 illustrates the day-by-day progression of corneal epithelium generation over 15 simulated days. Starting with only LESCs, tear film, and EpBM, the simulation shows basal cells proliferating and migrating centripetally toward the central cornea during the first few days. Wing cells emerge as intermediaries, facilitating layered structure formation. By day 7, a complete stratified structure forms with all cell types present at the appropriate positions, and by day 15, the tissue achieves a stable homeostatic state with consistent layer organization and thickness.

This timeline is particularly noteworthy considering the model implements biologically constrained cell cycle times with a minimum doubling interval of 8 hours under optimal conditions. Previous simulations used corneal epithelial cell doubling times ranging from 6 hours to 16 days [29]. Our model consistently produces tissue-level organization within 7–14 days, aligning with known corneal epithelial renewal periods observed *in vivo* [90].

### 3.2. Long-term stability and tissue architecture

Our simulation demonstrates long-term stability when run with calibrated parameters (detailed in S8–S11 Tables). Fig 7 shows the maintenance of consistent cell population dynamics across multiple simulation replicates over six months of

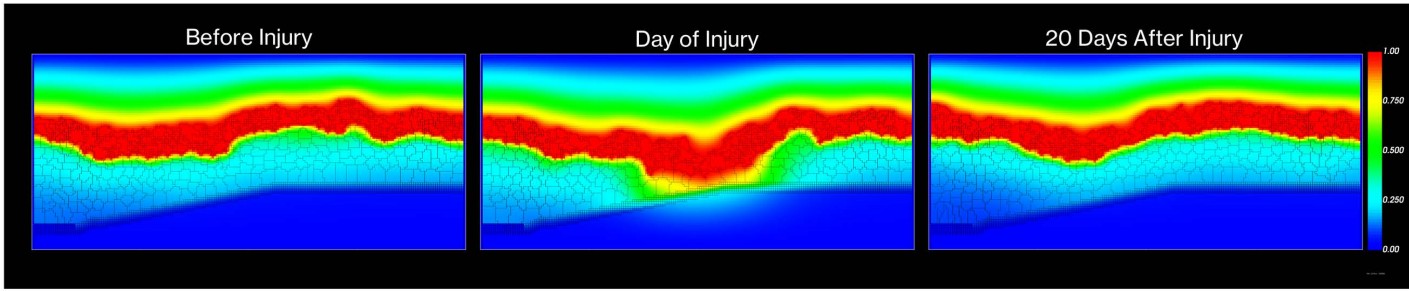

**Fig 5. Spatiotemporal evolution of the EGF concentration field. (A)** Baseline physiological gradient where the Tear film acts as a constant source (red) diffusing into the tissue. **(B)** Immediate post-injury state showing increased EGF influx through the epithelial breach. **(C)** Restored gradient 20 days post-injury following barrier re-establishment. Color scale indicates normalized EGF concentration from 0.0 (blue) to 1.0 (red).

**Table 1. Summary of core model components and rules.**

| Model quantity/ rule | Value (simulation units) | Approximate biological value/ target | Role in model/ interpretation |
|---|---|---|---|
| **Spatiotemporal Scales** | | | |
| **Lattice unit edge length** | 1 voxel unit | 2 μm | Spatial resolution of the epithelial cross-section and wound geometry. |
| **Lattice unit area** | 1 voxel$^2$ | 4.0 μm$^2$ | Area per lattice site directly derived from edge length. |
| **Time step** | 1 MCS | 6.0 min | Decision based on computing power and resolution of cellular behavior like movement and division. |
| **Simulation domain** | 200 × 90 voxels | 400 × 180 μm | Represents limbus + peripheral cornea. (Fig 3) while computationally tractable. |
| **Growth and Turnover** | | | |
| **Growth rate of proliferating cells** | Rate-limiting function (Eq. 3) | $\delta_i$ set so that, under high EGF and low crowding, cells can double volume in ≈ 8 h [29] | Combines biochemical (EGF) and mechanical (density/pressure) cues; defines maximum proliferative capacity under optimal conditions. |
| **EGF-dependent growth rate of cells** | Hill function of local EGF (Eq. 1) | Increase in growth when EGF exceeds a half-max level | Represents MAPK/ERK-like ultrasensitive growth-factor response and noise-filtering behavior [54,88]. |
| **Half-Max EGF concentration for EGF-dependent growth rate (Stem)/ (Basal)** | $k_{m,EGF,STEM} = 3.5$ arbritary units/ $k_{m,EGF,BASAL} = 7.0$ arbritary units | Local cell response at which growth rate is increased by ~50% | Calibrated so epithelial thickness and limbal–peripheral density gradients remain physiological while maintaining 7–14 day renewal [3]. |
| **Density inhibition rate of proliferating cells** | Hill function of effective pressure (Eq. 2) | Steep decline of growth with increasing crowding | Represents contact inhibition of proliferation in crowded epithelia [30]. |
| **Half-Max pressure for contact inhibition of cell growth** | $k_{m, density} = 125.0$ arbitrary units | Local pressure at which growth is reduced by ~50% | Fitted density/crowding level where proliferation becomes strongly inhibited at steady state. |
| **Superficial cell sloughing rate** | Stochastic Rule/Process (Eq. 8) | ≈ 1.75 days per layer → 7–14 days total renewal | Probabilistic removal of superficial cells at the tear interface; matches reported corneal epithelial turnover times [90,95]. |
| **Differentiation and EGF Dynamics** | | | |
| **Differentiation rules (LESC→Basal, Basal→Wing, Wing→Superficial)** | Logical rules based on neighbor set N and EpBM contact (Eqs. 4–6) | Immediate state transitions once spatial conditions are met | Implement deterministic, spatially triggered differentiation cascade from limbal niche to apical surface [45,48,49,57,58]. |
| **EGF reaction–diffusion dynamics** | Global diffusion rate = $186.0 \frac{voxel^2}{MCS}$ Global decay = $0.5$ MCS$^{-1}$ (Eq. 9) | $2.1 \times 10^{-8}$ cm$^2$/s [93] ~42–114 min [94] | Captures EGF gradients driven by a constant Tear Film source [34,35] and cell-type–dependent diffusion [50], regulating proliferation and migration during homeostasis and injury. |

Full mathematical details and parameter values are provided in S1–S5 Text.

simulated time. The frequency distribution of cell counts exhibits a bell-shaped curve, reflecting natural biological variability within controlled parameters. Q-Q analysis confirms that cell count distributions align closely with normal distributions, demonstrating the reproducibility and statistical reliability of our model's homeostatic state.

We validate our model's biological accuracy through two complementary approaches for measuring tissue thickness. The Center of Mass Difference method calculates the distance between the deepest cell layer (basal and stem cells) and the superficial cell layer, providing a direct measurement of epithelial thickness. This method shows that the tissue stabilizes at approximately 50 μm, which lies within the range of central corneal epithelial thickness reported in imaging studies

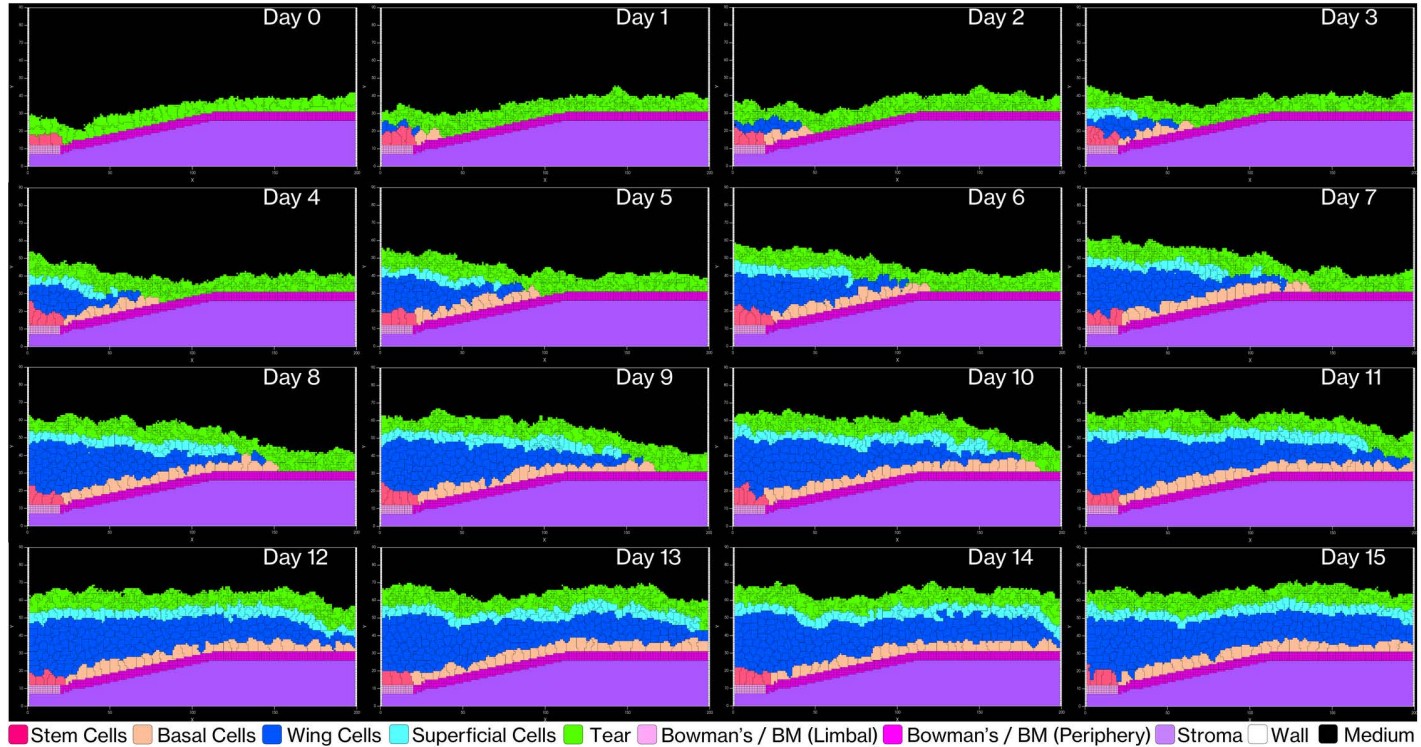

**Fig 6. Temporal progression of corneal epithelium development.** Time series (Day 0–15) showing the emergence of stratified layers from an initial population of Limbal Stem Cells (rose-red) and Basal cells (peach-orange). Wing cells (blue) and Superficial cells (cyan) appear via differentiation and migration, forming a stable homeostatic architecture by Day 15.

of healthy adults (roughly 48–55 μm). Limbal epithelial thickness in these same populations is consistently higher, with mean values around 80–85 μm. [60,83,84].

To assess regional variations, we employ Relative Position Tracking using the segmentation approach shown in Fig 8. By dividing the tissue into ten equal segments along the X-axis (0–200 μm) from limbus to periphery, we track the positions of superficial cells within each segment over six months of simulated time. This analysis reveals remarkable consistency across all segments, with thickness variations showing a standard deviation of only 2 μm, indicating highly uniform tissue architecture. Our measurements also demonstrate a characteristic thickness gradient from limbus to periphery, further validating our model's ability to maintain physiologically relevant tissue architecture.

### 3.3. CellTurnover dynamics

V-Cornea accurately captures the differential cell turnover rates observed in biological corneal epithelium. Peripheral corneal regions exhibit faster turnover, with nearly complete cellular substitution occurring approximately every seven days, matching literature reports that indicate corneal epithelium renewal every 7–10 days [95–97]. In contrast, the limbus region demonstrates a more gradual renewal rate, aligning with a 14-day turnover period.

This differential turnover pattern reflects natural corneal epithelial function, where limbal stem cells serve as a continuous source for epithelial regeneration while maintaining their own population through controlled division rates [7,90]. We maintain this realistic dynamic in our model through a balanced equilibrium between cell proliferation and death, even over extended simulation periods. This temporal heterogeneity emerges naturally from our model despite implementing

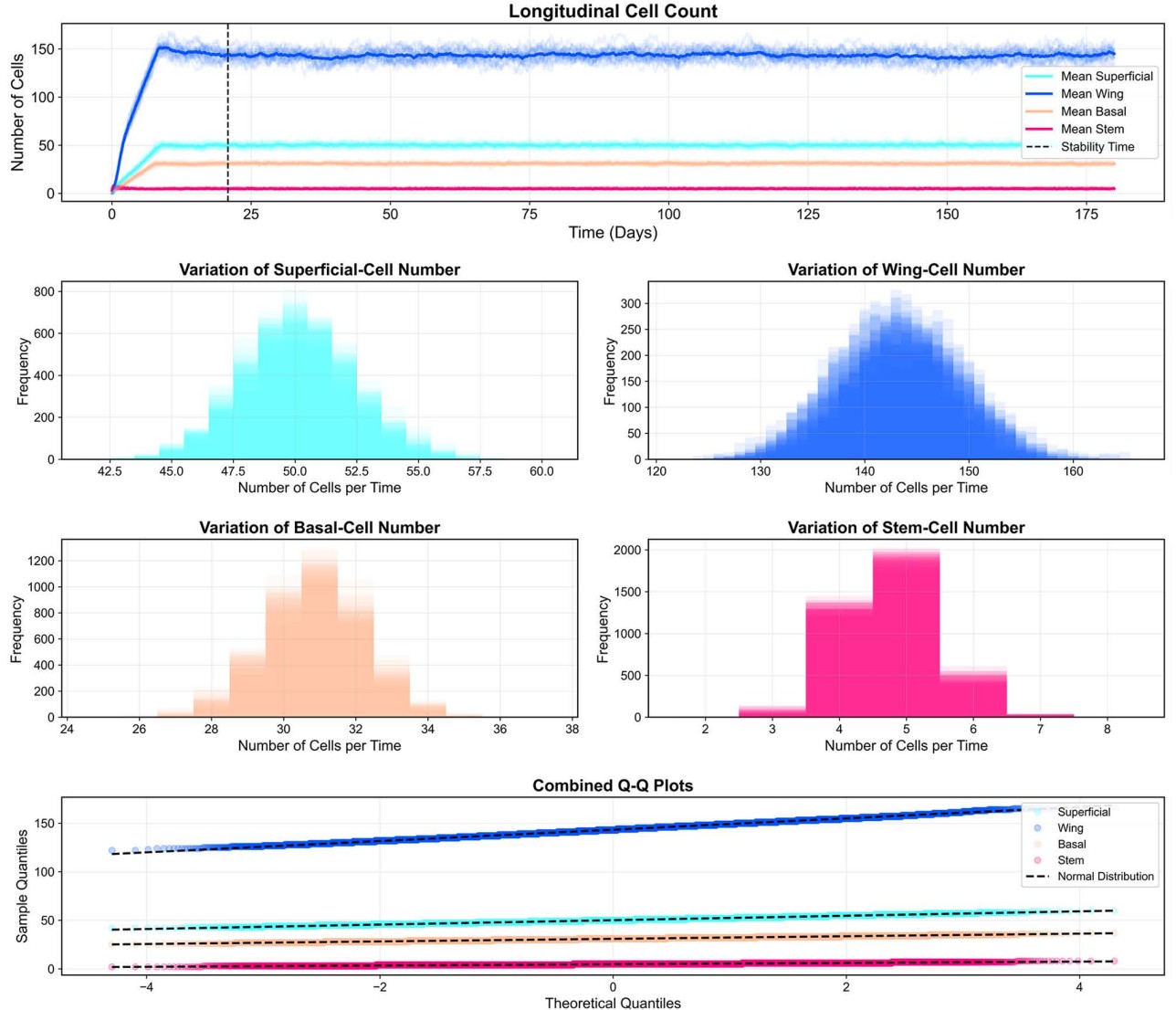

**Fig 7. Longitudinal quantitative analysis of cell population stability.** (Top) Mean cell counts for Superficial (cyan), Wing (blue), Basal (peach), and Stem (pink) cells over six months. The vertical dashed line marks stability from where data analysis begins. (Middle) Frequency distribution histograms of cell counts across replicates. (Bottom) Combined Q-Q plots demonstrating normality of cell count distributions compared to theoretical normal distributions (dashed lines).

consistent rules across all cells, demonstrating how V-Cornea captures complex tissue behaviors through simple cellular mechanisms.

### 3.4. Injury simulation and recovery

To evaluate our model's capacity to simulate tissue resilience and recovery, we introduced injuries of varying depths based on the classification system established by Scott et al. (2010) [3]. Our post-injury simulations revealed distinct recovery patterns that varied with injury depth, as illustrated in Fig 9.

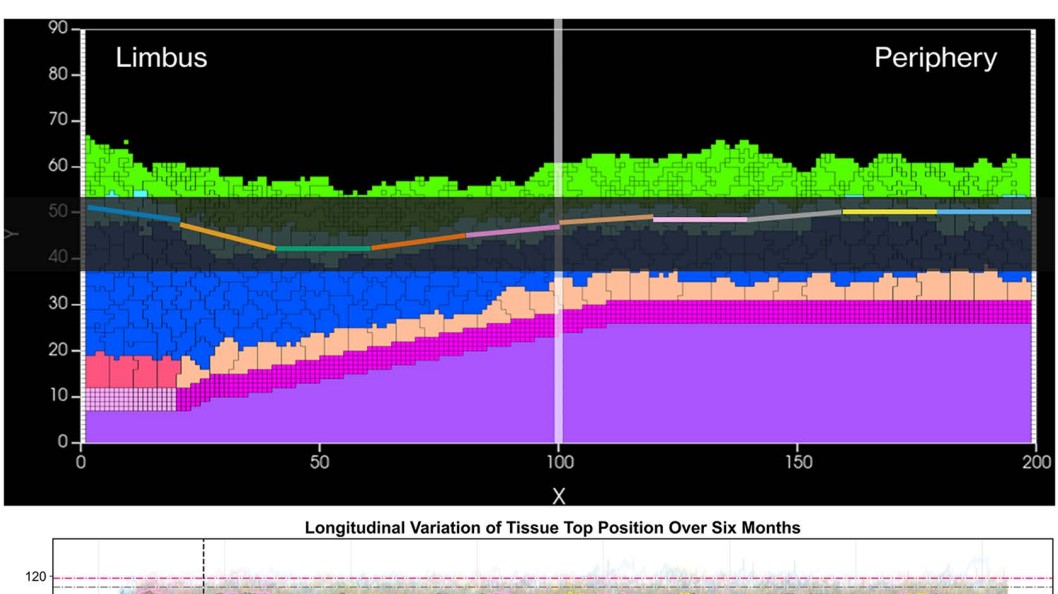

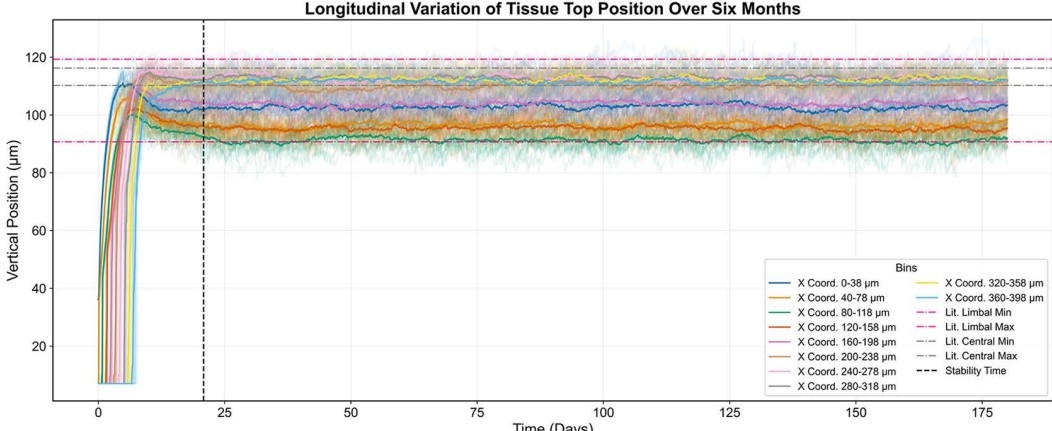

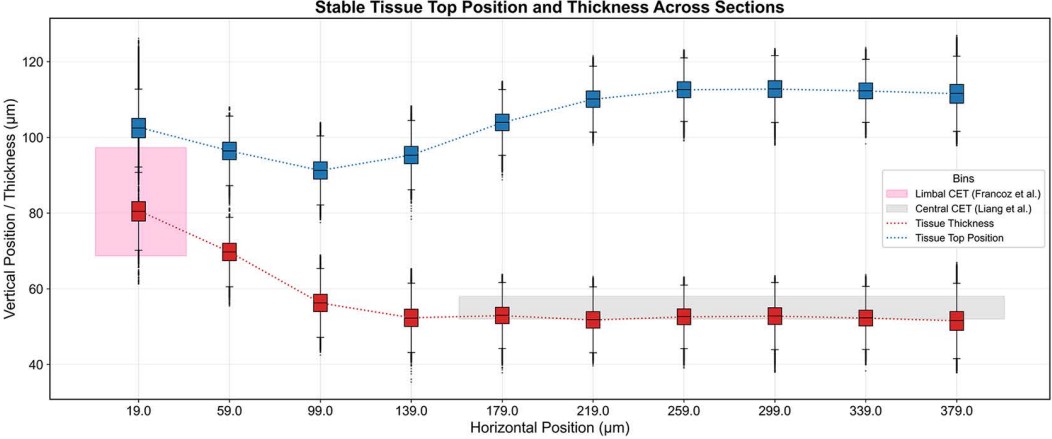

**Fig 8. Spatial segmentation analysis of tissue thickness and stability.** (Top) Cross-section showing the division of the tissue into ten X-axis segments (0–200 μm). (Middle) Longitudinal variation of tissue top position tracked per spatial bin. Colored lines represent simulated apical surface position. Dashed lines indicate literature ranges for Limbal (83.0 ± 14.3 μm; [84], Pink) and Central (55.0 ± 3.0 μm; [83], Gray) top positions. (Bottom) Spatial distribution of the epithelium at stability from Limbus (Bin 0) to Peripheral cornea (Bin 9). Blue box plots: Simulated Tissue Top Position. Red box plots: Simulated Tissue Thickness. Shaded areas indicate literature reference ranges for Limbal (Pink) and Central (Gray) thickness. Transition zones (Bins 1–3) are unshaded.

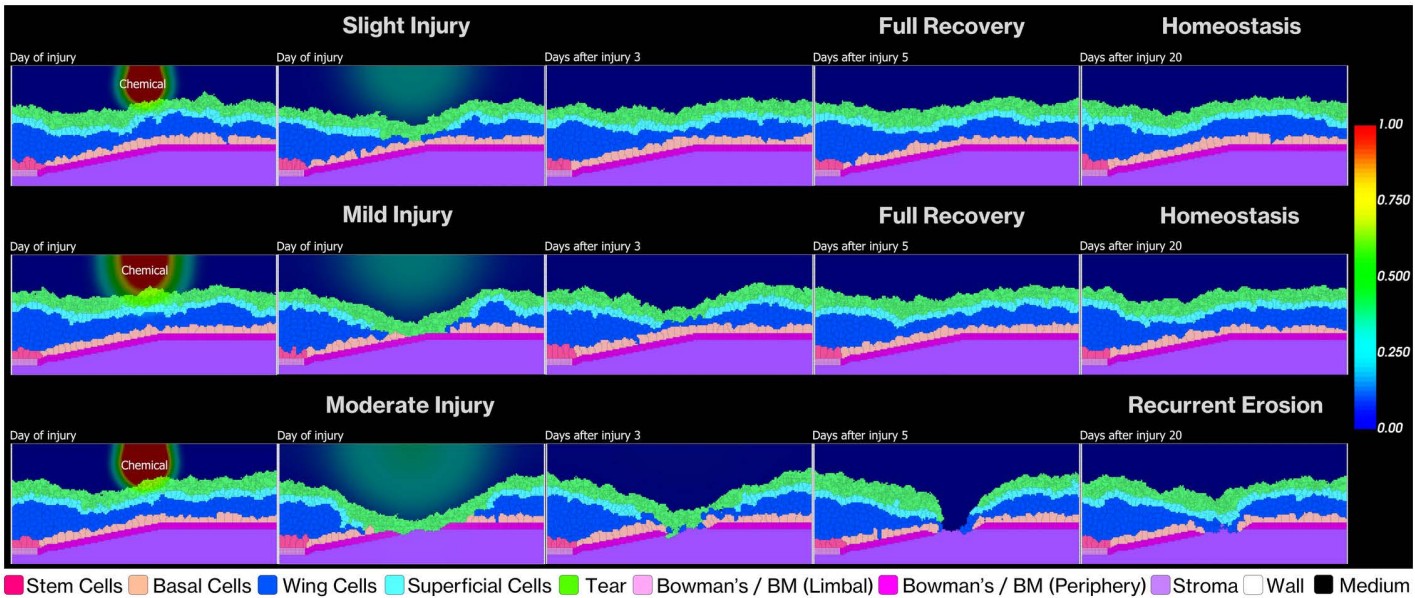

**Fig 9. Temporal progression of recovery following chemical injuries.** Rows display injury severity: Slight (750 arb. units), Mild (1500 arb. units), and Moderate (2500 arb. units). Columns show the timeline from injury to Day 20. Slight and Mild injuries (top/middle) achieve complete re-epithelialization and homeostasis. Moderate injury (bottom) compromising the basement membrane results in recurrent erosion and persistent epithelial defects.

For slight and mild injuries limited to the epithelial layers, we observed complete epithelial closure within 3–5 days post-injury. The simulation captured the immediate response to cell loss—increased EGF concentration within the epithelium that promoted basal cell proliferation and facilitated rapid wound closure. This timeline aligns with clinical studies showing that superficial corneal abrasions in humans typically resolve within 48–72 hours [7,98].

In contrast, moderate injuries that extended into the stroma and compromised the basement membrane produced markedly different outcomes. These deeper injuries resulted in incomplete recovery with persistent disruptions in the epithelial layer, resembling clinical patterns of recurrent corneal erosions (RCE). Fig 10 quantitatively demonstrates this difference—slight and mild injuries show cell populations returning to pre-injury levels within 5–7 days with stable homeostasis thereafter, while moderate injuries lead to sustained perturbations in wing and superficial cell populations over six months of simulation.

This emergent RCE-like behavior stems from a specific mechanism in our model. When the basement membrane becomes disrupted, basal epithelial cells fail to establish stable adhesion to the underlying stroma, resulting in repeated epithelial detachment after initial closure. The effect arises because the basement membrane serves dual critical functions: providing a substrate for epithelial cell adhesion through hemidesmosomes, while simultaneously maintaining basal cells in an undifferentiated state [48]. When basal cells lose contact with the basement membrane, they undergo premature terminal differentiation, compromising epithelial integrity.

### 3.5. Model validation and relationship to clinical conditions

The alignment between our simulation results and empirical data serves as validation of the model's biological relevance. The recovery period of 3–5 days for superficial injuries parallels findings from multiple clinical studies [7,99] and is consistent with the natural turnover rate of corneal epithelial cells [90].

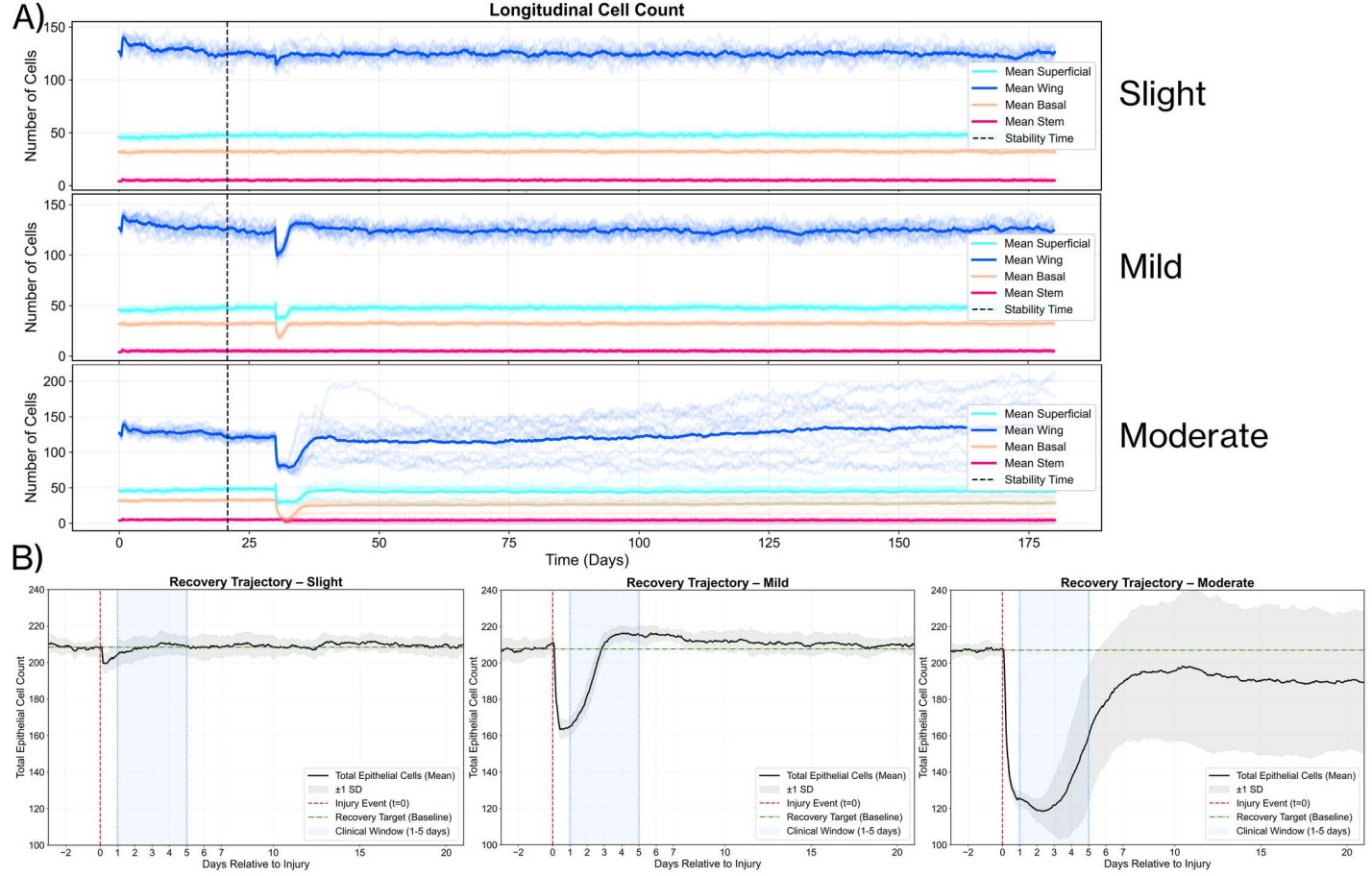

**Fig 10. Comparative quantitative analysis of post-injury cell dynamics. (A)** Longitudinal cell counts tracked over 6 months for slight (top), mild (middle), and moderate (bottom) injuries. Solid lines show mean counts for Superficial (cyan), Wing (blue), Basal (peach-orange), and Stem (rose-red) cells; shaded areas indicate standard deviation. Vertical dashed lines mark stability and injury time points. **(B)** Recovery trajectories of total epithelial cell counts relative to the injury event (t = 0). The black line represents the mean total count (± SD). The horizontal green dash-dot line indicates the pre-injury recovery target. The blue vertical band (1–5 days) delineates the clinically expected healing window; Moderate injury fails to return to baseline within this window.

More significant is our model's ability to reproduce key aspects of recurrent corneal erosions (RCE) despite not explicitly programming this pathology. In clinical RCE, basement membrane damage disrupts hemidesmosome formation and adhesion complexes, leading to persistent epithelial defects [72,73,100]. The molecular basis for this disruption involves processes such as improper processing of laminin-5, a key basement membrane component essential for stable epithelial adhesion [101].

Our current model does not incorporate specific molecular mechanisms of basement membrane regeneration, which would involve coordinated interaction between basal epithelial cells and stromal keratocytes [90]. This limitation interestingly provides an unintended benefit—creating a system where we can study the consequences of impaired basement membrane function in isolation from repair mechanisms.

The emergence of RCE-like features highlights how fundamental the basement membrane is to corneal epithelial stability. While our findings cannot directly inform the pathogenesis of clinical RCE, our model demonstrates that compromised basement membrane integrity alone is sufficient to produce recurrent epithelial breakdown, independent of

other factors that may contribute to the clinical condition. This computational finding supplements existing literature on basement membrane dysfunction in corneal pathologies [102], offering a simplified system where the isolated effects of basement membrane compromise can be observed.

Our focus on depth-based injury patterns, rather than specific injury mechanisms, reflects the clinical observation that healing time correlates strongly with injury depth regardless of the initial insult type. This consistency suggests that our model captures fundamental biological processes governing corneal wound healing. Future iterations could incorporate mechanisms for basement membrane regeneration to better distinguish between model limitations and accurate biological representations, as well as targeted comparisons with chemical-specific clinical data to refine the model's predictive capabilities.

To quantitatively link V-Cornea to human biology, we constrained the model using the limited number of relevant and available experimentally measured tissue-level features and then assessed additional emergent behaviors. Cell-level parameters (target cell sizes, mechanical constraints, EGF diffusion and decay) were bound by literature values or conservative approximations (S8–S11 Tables). Within these bounds, we calibrated proliferation and sloughing rates to reproduce physiological epithelial turnover times of 7–14 days. We then evaluated independent emergent properties—epithelial thickness and limbal–central gradients, injury-specific healing times, and recurrent erosion-like instability after basement membrane damage—against clinical and experimental data (Figs 3 and 6–10). Table 2 summarizes this calibration–validation workflow; a more detailed version is provided in S12 Table.

## 4. Discussion

### 4.1. Interpretation of findings

V-Cornea demonstrates that complex corneal epithelial behavior can emerge from relatively simple biological rules. Previous modeling approaches have typically addressed isolated aspects of corneal dynamics—mechanical properties through FEA or wound healing through PDEs—whereas our integrated approach illuminates how cellular-level decisions collectively generate sophisticated tissue-level organization. The maintenance of tissue homeostasis over extended simulation periods provides compelling evidence that fundamental corneal epithelial mechanisms can be effectively captured through a limited set of core cellular rules.

**Table 2. Key corneal features used to calibrate and validate V-Cornea.**

| Feature | Experimental/ Clinical Value (Range, with Refs) | V-Cornea Prediction (This Study) | How Used in Model |
|---|---|---|---|
| Epithelial thickness and limbal–central gradient | Human corneal epithelium ≈48–55 µm thick [31,83,103], 5–7 cell layers, limbal epithelium ~83 µm [84,104,105]. | Stable 50–54 µm centrally; thicker (7–8 layers) at limbus (Figs 3 and 8). | **Emergent validation**: from cell growth, differentiation, and mechanical rules on physiologically bounded cells. |
| Epithelial turnover time | Complete renewal in ~7–14 days [31,106–108]. | Renewal of superficial layers in 7–14 days (Figs 6 and 7) | **Constrained calibration**: Turnover window guided tuning of proliferation and sloughing rates. |
| Healing* time, slight/mild epithelial injury (intact BM) | Small injuries heal 1–3 days; larger epithelial injuries heal 3–5 days [87,109–111]. | Closure in 3–5 days with recovery of thickness and cell counts (Figs 9 and 10). | **Secondary constraint and validation**: Parameter sets failing clinical closure times rejected; model recovery matched clinical healing qualitatively. |
| Healing* after BM damage (moderate injuries) | BM disruption leads to slower closure and recurrent corneal erosion (RCE) over weeks-months [72,109–112]. | No stable homeostasis by 20 days; recurrent epithelial breakdown like RCE (Figs 9 and 10). | **Qualitative emergent behavior**: Recurrent breakdown of epithelia emerges naturally, no specific parameters to confer instability. |

A more detailed description is provided in S12 Table.

*Healing refers to structural re-epithelialization and restoration of a continuous epithelial surface, as assessed by disappearance of fluorescein-stained defects and normalization of epithelial thickness in clinical imaging.

Strong alignment between simulated turnover rates and empirical observations [90,95–97,108] validates that these basic rules accurately reflect underlying biological processes. Particularly significant is the successful replication of the mechanistic relationship between basement membrane contact and cell differentiation [45,48], a key regulatory pathway previously uncaptured in computational models. This reproduction of both structural and functional aspects of corneal maintenance demonstrates that essential features of epithelial self-organization can be captured in a simplified rule set.

### 4.2. Response to injury and recovery mechanisms

The V-Cornea model provides valuable insights into mechanistic relationships between tissue damage and recovery by simulating differential healing responses based on injury severity. The successful replication of recovery times for slight and mild injuries validates our implementation of EGF-mediated healing mechanisms [4,5]. Particularly noteworthy is the emergent relationship between epithelial barrier disruption and accelerated healing through increased EGF diffusion—a phenomenon that arises naturally from basic cellular rules rather than explicit programming.

More significant, however, is the model's emergent behavior when simulating basement membrane disruption. What initially appeared as a limitation—the inability to fully heal moderate injuries—actually reveals valuable insights into pathological conditions such as recurrent corneal erosions [72,113,114]. These simulations demonstrate that V-Cornea can reproduce clinically relevant outcomes from relatively simple rule sets, making it a valuable platform for both theoretical and applied research. The model provides a framework that experimentalists and clinicians can use to investigate mechanisms, test hypotheses, and potentially develop therapeutic interventions for corneal pathologies without exclusive reliance on animal models or extensive clinical trials.

These findings highlight a crucial direction for future model development: incorporating the dynamic interplay between epithelial cells and keratocytes in basement membrane regeneration. Such enhancements would improve predictive capabilities for moderate injuries while potentially offering new insights into therapeutic approaches for recurrent corneal erosions. The emergent behaviors observed in V-Cornea demonstrate its potential as a versatile and extensible platform that bridges basic science and clinical applications in corneal research.

V-Cornea serves as a valuable bridge between experimental endpoint measurements and the dynamic progression of healing responses. Its potential applications extend to toxicology, where it could provide a computational framework for eye irritation testing by correlating patterns of initial cellular damage with distinct adverse outcomes. This approach could eventually complement existing testing methods, though significant validation work remains necessary before such applications can meaningfully contribute to regulatory decisions.

### 4.3. Limitations and future directions

Although V-Cornea effectively simulates acute corneal injury response, several biological mechanisms require implementation to enhance its relevance for accidental eye exposure scenarios. Our treatment of the basement membrane as a non-regenerative entity represents a key limitation in accurately predicting recovery from chemical exposures. After acute injury, epithelial cells and stromal keratocytes regenerate the basement membrane through coordinated action [72,101] - a process that, once implemented, would significantly improve predictions for injuries compromising basement membrane integrity.

The current model's simplified representation of chemical injury mechanisms presents a second limitation. Chemical exposures are currently modeled using generic diffusion patterns and concentration-dependent cell death thresholds, without incorporating chemical-specific modes of action or cellular selectivity. While this approach successfully distinguishes between injury classifications based on depth, it fails to capture the diversity of chemical-tissue interactions observed in real exposures. Different chemicals (surfactants, acids, alkalis, organic solvents) interact with corneal tissue through distinct mechanisms - disrupting cell membranes, denaturing proteins, or altering cell metabolism. Crucially, real exposures involve complex paracrine interactions, where primary damage to superficial cells triggers secondary stress responses in neighboring, non-exposed cells through the release of inflammatory mediators and damage-associated

molecular patterns (DAMPs) Future iterations would benefit from experimental data on specific chemical biochemical binding affinity, potency, and cellular targets to better simulate how different chemical classes affect corneal tissue, particularly for consumer products containing multiple ingredients with varying modes of action.

A third critical limitation lies in the simplified representation of tear film dynamics. The current model assumes constant tear production and EGF levels, whereas recent research shows these factors change rapidly following acute chemical exposure [33]. Since tear composition significantly influences both initial chemical exposure and subsequent healing response, future versions should incorporate dynamic tear film properties, such as reflex tearing (which alters chemical dilution) and changes in diffusion rates caused by tear film instability.

The exclusion of stromal and endothelial layers also limits comprehensive assessment of chemical injury responses, particularly for exposures penetrating beyond the epithelium. Without these deeper layers, important acute responses such as stromal keratocyte activation and myofibroblast transformation [7] remain uncaptured. Expanding the model to include these components would enable more comprehensive simulation of tissue responses to chemical exposures of varying severity.

Emerging experimental technologies offer opportunities to significantly advance V-Cornea's biological relevance. Spatial single-cell RNA-sequencing data collected at different timepoints during acute injury and recovery could reveal critical temporal dynamics of cellular responses and identify key molecular players in wound healing. Such data would enable more sophisticated modeling of cell-state transitions and intercellular signaling networks, particularly for chemical exposures, providing deeper insights into adverse outcome pathways.

Future development priorities for V-Cornea should focus on implementing basement membrane regeneration mechanisms, developing dynamic models of tear film composition, integrating stromal and endothelial tissue layers, and incorporating inflammatory responses. The modular CompuCell3D implementation also provides a flexible framework that researchers can adapt for diverse corneal studies beyond acute chemical exposure. Collaboration with research groups across toxicology, ophthalmology, and basic corneal biology will strengthen V-Cornea's utility while advancing our broader understanding of corneal biology and pathology.

In addition, V-Cornea is currently implemented as a 2D radial cross-section rather than a full 3D representation of the corneal surface. As discussed in Section 2.2, this dimensional reduction allows us to capture local depth- and radius-dependent behaviors but cannot represent circumferential phenomena such as sectoral limbal failure or global patterns of conjunctivalization.

## 5. Conclusion

V-Cornea effectively simulates complex corneal epithelial behavior using an agent-based modeling approach with biologically-inspired rules. The model successfully reproduces key aspects of corneal epithelial dynamics, including tissue self-organization, maintenance of stable architecture, and physiologically relevant cell turnover rates. Validation against empirical measurements [90,95–97,108], establishes V-Cornea's ability to capture fundamental homeostatic and recovery processes in corneal epithelium.

The model's ability to predict differential healing responses based on injury depth represents a particularly significant achievement. V-Cornea demonstrates potential for investigating pathological conditions [72,113,114] through accurate reproduction of healing times for slight and mild injuries [4,5,7], coupled with emergent behavior mimicking recurrent corneal erosions when basement membrane disruption occurs. These capabilities emerge naturally from basic cellular rules, indicating that the model captures essential features of corneal wound healing dynamics.

Through its modular CompuCell3D implementation, V-Cornea provides an extensible framework that researchers can adapt for diverse applications in corneal biology. While the current version does not yet include basement membrane regeneration mechanisms and deeper tissue responses, adding these components represents a priority for future development. We invite collaboration with research groups interested in applying or extending the model's capabilities for

studying basic corneal biology, evaluating chemical toxicant effects, or exploring other aspects of corneal pathology. This open, collaborative approach aims to advance understanding of corneal biology while supporting the development of animal-free methods for assessing corneal responses to injury.

### Declaration of generative AI and AI-assisted technologies in the writing process

During the preparation of this work, the authors used OpenAI ChatGPT o1 and Anthropic Claude 3.7 Sonnet to improve readability and avoid repetition. After using these tools, the authors reviewed and edited the content as needed and take full responsibility for the content of the published article.

### Supporting information

**S1 Text. V-Cornea Supplemental Mathematical Formulation for Growth and Mitosis.** Detailed description of the equations governing EGF-dependent growth, density-dependent inhibition (Hill functions), and the volume-threshold mitosis rules used to control cell proliferation.
(DOCX)

**S2 Text. V-Cornea Supplemental Mathematical Formulation for Differentiation Rules.** Mathematical formulation of the state transitions between cell types, including Stem-to-Basal, Basal-to-Wing, and Wing-to-Superficial differentiation based on neighbor contact and spatial context.
(DOCX)

**S3 Text. V-Cornea Supplemental Mathematical Formulation for Cellular Movement and Mechanics.** Definition of the Cellular Potts Model Hamiltonian, including contact energies, Hookean link (spring) constraints, volume/surface constraints, and the chemotaxis algorithm used to control cell motility.
(DOCX)

**S4 Text. V-Cornea Supplemental Mathematical Formulation for Cell Death and Injury Implementation.** Cell Death and Injury Implementation. Mathematical formulation for natural cell turnover (sloughing), volume-based death, and the implementation of specific injury modes including mechanical ablation and chemical burns.
(DOCX)

**S5 Text. V-Cornea Supplemental Mathematical Formulation for EGF Reaction-Diffusion Dynamics.** Mathematical description of the spatiotemporal evolution of the Epidermal Growth Factor (EGF) field, including diffusion coefficients, decay rates, boundary conditions, and source/sink terms.
(DOCX)

**S1 Table. Stem cell behavioral rules and signal integration.** Summary of the functional behaviors (growth, mitosis, differentiation, movement) for Stem cells, detailing the mathematical forms, specific signals (EGF, Pressure), and associated model parameters.
(DOCX)

**S2 Table. Basal cell behavioral rules and signal integration.** Summary of the functional behaviors for Basal cells, including EGF-dependent growth, contact-mediated differentiation to Wing cells, and mechanical constraints.
(DOCX)

**S3 Table. Wing cell behavioral rules and signal integration.** Summary of the functional behaviors for Wing cells, focusing on the differentiation transition to Superficial cells and chemotactic movement.
(DOCX)

**S4 Table. Superficial cell behavioral rules and signal integration.** Summary of the functional behaviors for Superficial cells, highlighting unique mechanical constraints (Hookean links between cell centers for tension maintenance) and the probabilistic sloughing mechanism responsible for natural tissue turnover.
(DOCX)

**S5 Table. Boundary agent behaviors (Tear and Bowman's/EpBM).** Summary of the behaviors for non-cellular agents, including EGF secretion by Tear fluid and the vulnerability of Bowman's membrane/EpBM to chemical exposure.
(DOCX)

**S6 Table. Contact energy parameters matrix.** Matrix of contact energy values defining the adhesive strength between all pairs of cell types and environmental agents (Medium, Tear, Membranes, Stroma). These parameters govern differential adhesion, driving cell sorting and the maintenance of tissue stratification.
(DOCX)

**S7 Table. Chemical field definitions.** Description of the diffusive fields used in the model (EGF, Movement Bias, and Chemical Injury), including their biological roles, units, and transport processes.
(DOCX)

**S8 Table. Cell growth and mechanical constraint parameters.** Listing of the initial parameter values governing target volumes, surface areas, and growth regulation (Hill function coefficients) for all cell types (Stem, Basal, Wing, Superficial), with literature references.
(DOCX)

**S9 Table. Chemical field diffusion and decay parameters.** Parameter values for the diffusion coefficients of EGF, Chemical Injury (SLS), and Movement Bias across different cell types and tissue layers, including uptake and secretion rates.
(DOCX)

**S10 Table. Injury configuration and mechanical link parameters.** Settings for simulating injuries (ablation coordinates, chemical concentration/timing) and the Focal Point Plasticity (FPP) parameters used to model tension at the tissue boundaries.
(DOCX)

**S11 Table. Simulation controls and data logging flags.** List of boolean flags and settings used to enable/disable specific biological modules (mitosis, death) and configure output metrics (thickness plotting, cell counts) for debugging and analysis.
(DOCX)

**S12 Table. Model calibration and validation against clinical data.** A detailed comparison of V-Cornea model predictions versus experimental/clinical literature values for key physiological features (epithelial thickness, turnover time, healing rates) serving as the primary validation metrics.
(DOCX)

## Author contributions

**Conceptualization:** Joel Vanin, Michael Getz, Catherine Mahony, James A. Glazier.

**Data curation:** Michael Getz.

**Formal analysis:** Joel Vanin.

**Funding acquisition:** Catherine Mahony.

**Investigation:** Joel Vanin, Michael Getz, Catherine Mahony.

**Methodology:** Joel Vanin.

**Project administration:** Catherine Mahony, James A. Glazier.

**Resources:** Catherine Mahony.

**Software:** Joel Vanin.

**Supervision:** Catherine Mahony, Thomas B. Knudsen, James A. Glazier.

**Validation:** Joel Vanin.

**Visualization:** Joel Vanin.

**Writing – original draft:** Joel Vanin.

**Writing – review & editing:** Joel Vanin, Michael Getz, Catherine Mahony, Thomas B. Knudsen, James A. Glazier.

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
