## [Decision Letter · Decision Letter 0]

14 Nov 2025

PCOMPBIOL-D-25-01601

V-Cornea: A computational model of corneal epithelium homeostasis, injury, and recovery

PLOS Computational Biology

Dear Dr. Glazier,

Thank you for submitting your manuscript to PLOS Computational Biology. After careful consideration, we feel that it has merit but does not fully meet PLOS Computational Biology's publication criteria as it currently stands. Therefore, we invite you to submit a revised version of the manuscript that addresses the points raised during the review process.

We look forward to receiving your revised manuscript.

Kind regards,

David Basanta Gutierrez

Academic Editor

PLOS Computational Biology

Dimitrios Vavylonis

Section Editor

PLOS Computational Biology

**Journal Requirements:**

1) Please provide an Author Summary. This should appear in your manuscript between the Abstract (if applicable) and the Introduction, and should be 150-200 words long. The aim should be to make your findings accessible to a wide audience that includes both scientists and non-scientists. Sample summaries can be found on our website under Submission Guidelines:

3) We notice that your supplementary Tables, and information are included in the manuscript file. Please remove them and upload them with the file type 'Supporting Information'. Please ensure that each Supporting Information file has a legend listed in the manuscript after the references list.

Potential Copyright Issues:

i) Figure 1. Please confirm whether you drew the images / clip-art within the figure panels by hand. If you did not draw the images, please provide (a) a link to the source of the images or icons and their license / terms of use; or (b) written permission from the copyright holder to publish the images or icons under our CC BY 4.0 license. Alternatively, you may replace the images with open source alternatives. See these open source resources you may use to replace images / clip-art:

ii) Figure 3. We noted that you stated in the legend of the figure "Image cropped from Sorenson & Brelje, Atlas of Human Histology, 3rd Edition, 2014). Copyright 2014 T. Clark Brelje and Robert L. Sorenson."

Please provide written permission from the copyright holder/author to publish this under our CC BY 4.0 license, or remove the figure / replace the image.

You will find the content copyright permission form attached , please ask the owner of the figure to complete/sign it.

6) Please update the Data Availability statement in the online submission form to match the one mentioned in the manuscript. 

**Reviewers' comments:**

Reviewer's Responses to Questions

Reviewer #1: In this work the authors present V-Cornea, an agent-based computational model built in CompuCell3D to simulate corneal epithelial homeostasis and injury response, addressing limitations of current ocular irritation assessments, particularly in predicting long-term effects and recovery. The model integrates biologically inspired rules for cell behaviors and key signaling pathways such as EGF, enabling in vitro to in vivo extrapolation of tissue-level outcomes.

Overall, V-Cornea demonstrates potential as a virtual-tissue platform for toxicological testing, drug discovery, and therapeutic optimization across diverse corneal injury scenarios.

The paper would benefit from three minor revisions:

1) Add more comparisons between the model prediction and experimental data in a clear figure that demonstrates that the model recapitulates experimental data with physiological model parameters. Adding such a figure is crucial.

2) Add a detailed discussion on the limitations of the 2D modeling compared to a full 3D model (how do scaling relations change? how do injury progression changes? etc.)

3) It is not easy to get a clear, full picture of the equations and the biological numbers (and their justification) that are in many tables and sub-sections. Add a clear table with all the numbers with units and their corresponding equations.

Reviewer #2: This manuscript presents an agent-based model on corneal epithelial development dynamics, which combines cell movement driven by intercellular forces and cell differentiation governed by a set of simplified biological rules. The model simulation was implemented through the CompuCell3D framework. The model reproduces corneal epithelial homeostasis and responses to mild injuries. The model predictions for moderate injuries that harms the basement membrane recapitulates recurrent corneal erosions, although further incorporation of cellular dynamics beyond the epithelium is necessary for accurate quantitative prediction. The model provides a prototype predictive tool for biomedical and clinical studies of corneal injuries, which can be extended in the future with more cellular mechanisms incorporated. The manuscript is written with high clarity.

Major comments:

1. Is there any physiological or modeling reason behind the Hill exponent of 4 in the dependence of proliferation rates of stem cell and basal cell on EGF and cell density?

2. Where are the various layers in Fig.5? I presume the red region before injury and after recovery correspond to the tear film, as EGF concentration was highest there? What confuses me is the EGF concentration beyond that tear layer. Shouldn’t there be nothing beyond the tear? If it is a numeric convenience to assume diffusion of EGF beyond the tear film, would this artificial diffusion cause underestimation or some sort of distortion of the EGF level in the cornea?

Minor comments:

1. Ln. 114-119: This sentence is really difficult to read. To improve clarity, it would be better to break it into two sentences.

2. Ln. 257: “While … “ According to the meaning of the sentence, “As/Since/Because” may be more appropriate for the first word?

3. The “if” conditions given in Eqs. (4) and (6) contain math notations that I don’t understand.

4. What is the reason that the number of layers of wing cells is highest right above the stem cells and become largely constant towards the central cornea?

**Have the authors made all data and (if applicable) computational code underlying the findings in their manuscript fully available?**

Reviewer #1: None

Reviewer #2: Yes

PLOS authors have the option to publish the peer review history of their article (what does this mean? ). If published, this will include your full peer review and any attached files.

**Do you want your identity to be public for this peer review?** For information about this choice, including consent withdrawal, please see our Privacy Policy .

Reviewer #1: No

Reviewer #2: No

**Figure resubmission:**
---

## [Editor Report · Decision Letter 1]

12 Dec 2025

Dear Prof. Glazier,

We are pleased to inform you that your manuscript 'V-Cornea: A computational model of corneal epithelium homeostasis, injury, and recovery' has been provisionally accepted for publication in PLOS Computational Biology.

Best regards,

David Basanta Gutierrez

Academic Editor

PLOS Computational Biology

Dimitrios Vavylonis

Section Editor

PLOS Computational Biology

---

## [Editor Report · Acceptance letter]

PCOMPBIOL-D-25-01601R1

V-Cornea: A computational model of corneal epithelium homeostasis, injury, and recovery

Dear Dr Glazier,

I am pleased to inform you that your manuscript has been formally accepted for publication in PLOS Computational Biology. Your manuscript is now with our production department and you will be notified of the publication date in due course.

With kind regards,

Anita Estes
